# Dropout's Dream Land: Generalization from Learned Simulators to Reality

## Abstract

A World Model is a generative model used to simulate an environment. World Models have proven capable of learning spatial and temporal representations of Reinforcement Learning environments. In some cases, a World Model offers an agent the opportunity to learn entirely inside of its own dream environment. In this work we explore improving the generalization capabilities from dream environments to reality (Dream2Real). We present a general approach to improve a controller's ability to transfer from a neural network dream environment to reality at little additional cost. These improvements are gained by drawing on inspiration from domain randomization, where the basic idea is to randomize as much of a simulator as possible without fundamentally changing the task at hand. Generally, domain randomization assumes access to a pre-built simulator with configurable parameters but oftentimes this is not available. By training the World Model using dropout, the dream environment is capable of creating a nearly infinite number of *different* dream environments. Our experimental results show that Dropout's Dream Land is an effective technique to bridge the reality gap between dream environments and reality. Furthermore, we additionally perform an extensive set of ablation studies.

## 1 Introduction

Reinforcement learning (Sutton & Barto, 2018) (RL) has experienced a flurry of success in recent years, from learning to play Atari (Mnih et al., 2015) to achieving grandmaster level performance in StarCraft II (Vinyals et al., 2019). However, in all these examples, the target environment is a simulator that can be directly trained in. Reinforcement learning is often not a practical solution without a simulator of the environment.

Sometimes the target environment is expensive, dangerous, or even impossible to interact with. In these cases, the agent is trained in a simulated source environment. Approaches that train an agent in a simulated environment with the hopes of generalization to the target environment experience a common problem referred to as the *reality gap* (Jakobi et al., 1995). One approach to bridge the reality gap is domain randomization (Tobin et al., 2017). The basic idea is that an agent which can perform well in an ensemble of simulations will also generalize to the real environment (Antonova et al., 2017; Tobin et al., 2017; Mordatch et al., 2015; Sadeghi & Levine, 2016). The ensemble of simulations is generally created by randomizing as much of the simulator as possible without fundamentally changing the task at hand. Unfortunately, this approach is only applicable when a simulator is provided and the simulator is configurable.

A recently growing field, World Models (Ha & Schmidhuber, 2018), focuses on the side of this problem when the simulation does not exist. World Models offer a general framework for optimizing controllers directly in *learned* simulated environments. The learned dynamics model can be viewed as the agent's dream environment. This is an interesting area because it removes the need for an agent to operate in the target environment. Some related approaches (Łukasz Kaiser et al., 2020; Hafner et al., 2019; 2020; Sekar et al., 2020; Sutton, 1990; Kurutach et al., 2018) focus on an adjacent problem which allows the controller to continually interact with the target environment.

Despite the recent improvements (Łukasz Kaiser et al., 2020; Hafner et al., 2019; Sekar et al., 2020; Kim et al., 2020; Hafner et al., 2020) of World Models, none of them address the issue that World Models are susceptible to the reality gap. The learned dream environment can be viewed as the source domain and the true environment as the target domain. Whenever there are discrepancies between the source and target domains the reality gap can cause problems. Even though World Models

suffer from the reality gap, none of the domain randomization approaches are directly applicable because the dream environment does not have easily configurable parameters.

In this work we present Dropout's Dream Land (DDL), a simple approach to bridge the reality gap from learned dream environments to reality. Dropout's Dream Land was inspired by the first principles of domain randomization, namely, train a controller on a large set of *different* simulators which all adhere to the fundamental task of the target environment. We are able to generate a nearly infinite number of different simulators via the insight that dropout (Srivastava et al., 2014) can be understood as learning an ensemble of neural networks (Baldi & Sadowski, 2013).

Our empirical results demonstrate the advantage of Dropout's Dream Land over baseline (Ha & Schmidhuber, 2018; Kim et al., 2020) approaches. Furthermore, we perform an extensive set of ablation studies which indicate the source of generalization improvements, requirements for the method to work, and when the method is most useful.

## 2 RELATED WORKS

### 2.1 DROPOUT

Dropout (Srivastava et al., 2014) was introduced as a regularization technique for feedforward and convolutional neural networks. In its most general form, each unit is dropped with a probability $p$ during the training process. Recurrent neural networks (RNNs) initially had issues benefiting from Dropout. Zaremba et al. (2014) suggests not to apply dropout to the hidden state units of the RNN cell. Gal & Ghahramani (2016b) shortly after show that the mask can also be applied to the hidden state units, but the mask must be fixed across the sequence during training.

In this work, we follow the dropout approach from Gal & Ghahramani (2016b) when training the RNN. More formally, for each sequence, the boolean masks $\mathbf{m}_{xi}$, $\mathbf{m}_{xf}$, $\mathbf{m}_{xw}$, $\mathbf{m}_{xo}$, $\mathbf{m}_{hi}$, $\mathbf{m}_{hf}$, $\mathbf{m}_{hw}$, and $\mathbf{m}_{ho}$ are sampled, then used in the following LSTM update:

$$\mathbf{i}_t = \mathbf{W}_{xi}(\mathbf{x}_t \odot \mathbf{m}_{xi}) + \mathbf{W}_{hi}(\mathbf{h}_{t-1} \odot \mathbf{m}_{hi}) + \mathbf{b}_i, \tag{1}$$

$$\mathbf{f}_t = \mathbf{W}_{xf}(\mathbf{x}_t \odot \mathbf{m}_{xf}) + \mathbf{W}_{hf}(\mathbf{h}_{t-1} \odot \mathbf{m}_{hf}) + \mathbf{b}_f, \tag{2}$$

$$\mathbf{w}_t = \mathbf{W}_{xw}(\mathbf{x}_t \odot \mathbf{m}_{xw}) + \mathbf{W}_{hw}(\mathbf{h}_{t-1} \odot \mathbf{m}_{hw}) + \mathbf{b}_w, \tag{3}$$

$$\mathbf{o}_t = \mathbf{W}_{xo}(\mathbf{x}_t \odot \mathbf{m}_{xo}) + \mathbf{W}_{ho}(\mathbf{h}_{t-1} \odot \mathbf{m}_{ho}) + \mathbf{b}_o, \tag{4}$$

where $\mathbf{x}_t$, $\mathbf{h}_t$, and $\mathbf{c}_t$ are the input, hidden state, and cell state, respectively, $\mathbf{W}_{xi}$, $\mathbf{W}_{xf}$, $\mathbf{W}_{xw}$, $\mathbf{W}_{xo} \in \mathbb{R}^{d \times r}$ $\mathbf{W}_{hi}$, $\mathbf{W}_{hf}$, $\mathbf{W}_{hw}$, $\mathbf{W}_{ho} \in \mathbb{R}^{d \times d}$ are the LSTM weight matrices, and $\mathbf{b}_i$, $\mathbf{b}_f$, $\mathbf{b}_w$, $\mathbf{b}_o \in \mathbb{R}^d$ are the LSTM biases. The masks are fixed for the entire sequence, but may differ between sequences in the mini-batch.

### 2.2 DOMAIN RANDOMIZATION

The goal of domain randomization (Tobin et al., 2017; Sadeghi & Levine, 2016) is to create many different versions of the dynamics model with the hope that a policy generalizing to all versions of the dynamics model will do well on the true environment. Figure 1 illustrates many simulated environments ($\hat{e}^j$) overlapping with the actual environment ($e^*$). Simulated environments are often far cheaper to operate in than the actual environment. Hence, it is desirable to be able to perform the majority of interactions in the simulated environments.

Randomization has been applied on observations (e.g., lighting, textures) to perform robotic grasping (Tobin et al., 2017) and collision avoidance of drones (Sadeghi & Levine, 2016). Randomization has also proven useful when applied to the underlying dynamics of simulators (Peng et al., 2018). Often, both the observations and simulation dynamics are randomized (Andrychowicz et al., 2020).

Domain randomization generally uses some pre-existing simulator which then injects randomness into specific aspects of the simulator (e.g., color textures, friction coefficients). Each of the simulated environments in Figure 1 can be thought of as a noisy sample of the pre-existing simulator. To the best of our knowledge, domain randomization has yet to be applied to entirely learned simulators.

### 2.3 WORLD MODELS

The world model (Ha & Schmidhuber, 2018) has three modules trained separately: (i) vision module ($V$); (ii) dynamics module ($M$); and (iii) controller ($C$). A high-level view is shown in Algorithm 1.

**Algorithm 1** World Models: Training in dreams.
1: Initialize parameters of $V$, $M$, and $C$
2: Collect $N$ trajectories $\mathbf{o}$, $d$, and $\mathbf{a}$ from $e^*$
3: Optimize $V$ on observations $\mathbf{o}$
4: Generate embeddings $\mathbf{z}$ for $\mathbf{o}$ with $V$
5: Optimize $M$ on $\mathbf{z}$ and $d$
6: Generate dream environment $\hat{e}$ from $M$
7: **for** iteration=1, 2, . . . **do**
8:     Optimize $C$ via interactions with $\hat{e}$

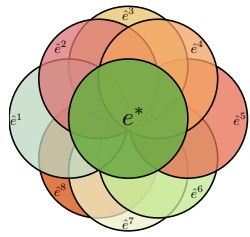

Figure 1: $e^*$ is the actual environment, and $\hat{e}^j$'s are randomized variants of the simulated environment.

The vision module ($V$) is a variational autoencoder (VAE) (Kingma & Welling, 2013), which maps image observation ($\mathbf{o}$) to a lower-dimensional representation $\mathbf{z} \in \mathbb{R}^n$.

The dynamics model ($M$) is a mixture density network recurrent neural network (MDN-RNN) (Ha & Schmidhuber, 2018; Graves, 2013). It is implemented as an LSTM followed by a fully-connected layer outputting parameters for a Gaussian mixture model with $k$ components. Each feature has $k$ different $\pi$ parameters for the logits of multinomial distribution, and $(\mu, \sigma)$ parameters for the $k$ components in the Gaussian mixture. At each timestep, the MDN-RNN takes in the state $\mathbf{z}$ and action $\mathbf{a}$ as inputs and predicts $\boldsymbol{\pi}, \boldsymbol{\mu}, \boldsymbol{\sigma}$. To draw a sample from the MDN-RNN, we first sample the multinomial distribution parameterized by $\boldsymbol{\pi}$, which indexes which of the $k$ normal distributions in the Gaussian mixture to sample from. This is then repeated for each of the $n$ features. Depending on the experiments, Ha & Schmidhuber (2018) also include an auxiliary head to the LSTM which predicts whether the episode terminates ($d$).

The controller ($C$) is responsible for deciding what actions to take. It takes features produced by the encoder $V$ and dynamics model $M$ as input (not the raw observations). The simple controller is a single-layer model which uses an evolutionary algorithm (CMA-ES (Hansen & Ostermeier, 2001)) to find its parameters. Depending on the problem setting, the controller ($C$) can either be optimized directly on the target environment ($e^*$) or on the dream environment ($\hat{e}$). This paper is focused on the case of optimizing exclusively in the dream environment.

## 3    Dropout's Dream Land

In this work we introduce Dropout's Dream Land (DDL). Dropout's Dream Land is the first work to offer a strategy to bridge the *reality gap* between learned neural network dynamics models and reality. Traditional domain randomization generates many *different* dynamics models by randomizing configurable parameters of a given simulation. This approach does not apply to neural network dynamics models because they generally do not have configurable parameters (such as textures and friction coefficients). In Dropout's Dream Land, the controller can interact with billions[1] of dream environments, whereas previous works (Ha & Schmidhuber, 2018; Kim et al., 2020) only use one dream environment. A naive way to go about this would be to train a large collection of neural network world models. However, this would be computationally expensive.

To keep the computational cost low, we go about this by applying dropout to the dynamics model in order to form different dynamics models. Crucially, dropout is applied at **both** training time and inference time of $M$. Each unique dropout mask applied to the dynamics model $M$ can be viewed as a different environment. Similar to the spirit of domain randomization, an agent is expected to perform well in the real environment if it can perform well in all the different simulated environments.

### 3.1    Learning the Dream Environment

The Dropout's Dream Land environments are built around the dynamics model $M$. During training time the controller interactions are described by Figure 2. In Figure 2 $\hat{r}$, $\hat{d}$, and $\hat{\mathbf{z}}$ are generated entirely by $M$. In this work $M$ is an LSTM where $\mathbf{x} = [\mathbf{z}^\top, \mathbf{a}^\top]^\top$ from equations (1)-(4). The LSTM is followed by multiple heads for predictions of the latent state ($\hat{\mathbf{z}}$), reward ($\hat{r}$) and termination ($\hat{d}$). The reward and termination heads are simple fully-connected layers. Latent state prediction is

---

[1]In practice we are bounded by the total number of steps instead of every possible environment.

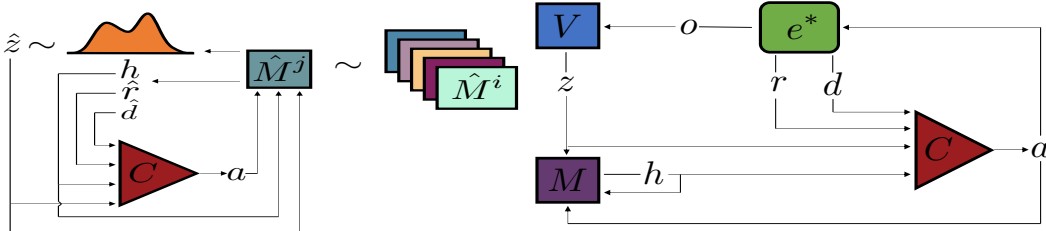

Figure 2: Interactions with the dream environment. A dropout mask is sampled at every step yielding a new $\hat{M}^j$.

Figure 3: Interactions with the real environment. The controller being optimized only interacts with the real environment during the final testing phase.

done with a MDN-RNN (Ha & Schmidhuber, 2018; Graves, 2013), but this could be replaced by any other neural network that supports dropout (e.g., GameGAN (Kim et al., 2020)).

### 3.1.1 LOSS FUNCTION

The dynamics model $M$ jointly optimizes all three heads where the loss of a single transition is defined as:

$$\mathcal{L}^M = \mathcal{L}^z + \alpha_r \mathcal{L}^r + \alpha_d \mathcal{L}^d. \tag{5}$$

Here, $\mathcal{L}^z = -\sum_{i=1}^{n} \log(\sum_{j=1}^{k} \hat{\pi}_{i,j} \mathcal{N}(z_i | \hat{\mu}_{i,j}, \hat{\sigma}_{i,j}^2))$ is a mixture density loss for the latent state predictions, where $n$ is the size of the latent feature vector $z$, $\hat{\pi}_{i,j}$ is the $j$th component probability for the $i$th feature, $\hat{\mu}_{i,j}, \hat{\sigma}_{i,j}$ are the corresponding mean and standard deviation. $\mathcal{L}^r = (r - \hat{r})^2$ is the square loss on rewards, where $r$ and $\hat{r}$ are the true and estimated rewards, respectively. $\mathcal{L}^d = -d \log(\hat{d}) - (1 - d) \log(1 - \hat{d})$ is the cross-entropy loss for termination prediction, where $d$ and $\hat{d}$ are the true and estimated probabilities of the episode ending, respectively. Constants $\alpha_d$ and $\alpha_r$ in (5) are for trading off importance of the termination and reward objectives. The loss ($\mathcal{L}^M$) is aggregated over each sequence and averaged across the mini-batch.

### 3.1.2 TRAINING DYNAMICS MODEL $M$ WITH DROPOUT

At training time of $M$ (Algorithm 1 Line 5), we apply dropout (Gal & Ghahramani, 2016b) to the LSTM to simulate different random environments. For each input and hidden unit, we first sample a boolean indicator with probability $p_{\text{train}}$. If the indicator is 1, the corresponding input/hidden unit is masked. Masks $\mathbf{m}_{xi}, \mathbf{m}_{xf}, \mathbf{m}_{xw}, \mathbf{m}_{xo}, \mathbf{m}_{hi}, \mathbf{m}_{hf}, \mathbf{m}_{hw}$, and $\mathbf{m}_{ho}$ are sampled independently (Equations (1)-(4)). When training the RNN, each mini-batch contains multiple sequences. Each sequence uses an independently sampled dropout mask. We fix the dropout mask for the entire sequence as this was previously found to be critically important (Gal & Ghahramani, 2016b).

Training the RNN with many different dropout masks is critical in order to generate multiple different dynamics models. At the core of domain randomization is the requirement that the randomizations do not fundamentally change the task. This constraint is violated if we do not train the RNN with dropout but apply dropout at inference time (explored further in Section 4.4). After optimizing the dynamics model $M$, we can use it to construct dream environments for controller training (Section 3.2).

In this work, we never sample masks to apply to the action (**a**). We do not zero out the action because in some environments this could imply the agent taking an action (ex: moving to the left). This design choice could be changed depending on the environment, for example, when a zero'd action corresponds to a no-op or a sticky action.

## 3.2 TRAINING THE CONTROLLER

### 3.2.1 INTERACTING WITH DROPOUT'S DREAM LAND

Interactions with the dream environment (Algorithm 1 Line 8) can be characterized as training time for the controller ($C$) and inference time of the dynamics model ($M$). An episode begins by generating the initial latent state vector $\hat{\mathbf{z}}$ by either sampling from a standard normal distribution or sampling from the starting points of the observed trajectories used to train $M$ (Ha & Schmidhuber, 2018). The hidden cell (**c**) and state (**h**) vectors are initialized with zeros.

The controller ($C$) decides the action to take based on $\hat{\mathbf{z}}$ and $\mathbf{h}$. In Figure 2 the controller also observes $\hat{r}$ and $\hat{d}$ but these are exclusively used for the optimization process of the controller. The controller then performs an action $\mathbf{a}$ on a dream environment.

A new dropout mask is sampled (with probability $p_{\text{infer}}$) and applied to $M$. We refer to the masked dynamics model as $M^j$ and the corresponding Dropout's Dream Land environment as $\hat{e}^j$. The current latent state $\hat{\mathbf{z}}$ and action $\mathbf{a}$ are concatenated, and passed to $M^j$ to perform a forward pass. The episode terminates based on a sample from a Bernoulli distribution parameterized by $\hat{d}$. The dream environment then outputs the latent state, LSTM's hidden state, reward, and whether the episode terminates.

It is crucial to apply dropout at inference time (of the dynamics model $M$) in order to create *different* versions of the dream environment for the controller $C$. In addition to randomizing masks at every step, we will also consider several other variants of how to apply dropout in Sections 4.3 and 4.4.

DDL's use of dropout is different from traditional applications of dropout. In related works (Kahn et al., 2017) Monte-Carlo (MC) Dropout (Gal & Ghahramani, 2016a) has been used to approximate the mean and variance of output predictions from an ensemble. We emphasize that DDL does not use MC Dropout. The purpose of DDL's approach to dropout is to generate many *different* versions of the dynamics model. More explicitly, the controller is trained to maximize expected returns across many different dynamics models in the ensemble as opposed maximizing expected returns on the ensemble average.

Dropout has also traditionally been used as a model regularizer. Dropout as a model regularizer is only applied at training time but not at inference time. The usual trade-off is lower test loss at the cost of higher training loss (Srivastava et al., 2014; Gal & Ghahramani, 2016b). However, DDL's ultimate goal is not lower test loss. In fact, we expect applying dropout at inference time will consistently make the test loss worse (this will be experimentally verified in Section 4.2). The ultimate goal is providing dream environments to a controller so that the optimal policy in Dropout's Dream Land also maximizes expected returns in the target environment ($e^*$).

### 3.2.2 TRAINING WITH CMA-ES

We follow the same controller optimization procedure as was done in World Models (Ha & Schmidhuber, 2018) and GameGAN (Kim et al., 2020) on their DoomTakeCover experiments. We train the controller with CMA-ES (Hansen & Ostermeier, 2001). Each controller in the population (of size $N_{\text{pop}}$) reports their mean returns on a set of $N_{\text{trials}}$ episodes generated in Section 3.2.1. As controllers in the population do not share a dream environment, the probability of controllers interacting with the same sequence of dropout masks is vanishingly small. Let $N_{\text{max\_ep\_len}}$ be the maximum number of steps in an episode. In a single CMA-ES iteration, the population as a whole can interact with $N_{\text{pop}} \times N_{\text{trials}} \times N_{\text{max\_ep\_len}}$ *different* environments. In our experiments, $N_{\text{pop}} = 64$, $N_{\text{trials}} = 16$, and $N_{\text{max\_ep\_len}}$ is 1000 for CarRacing and 2100 for DoomTakeCover. This potentially results in $> 1000000$ different environments at each generation.

### 3.2.3 DREAM LEADER BOARD

After every fixed number of generations (25 in our experiments), the best controller in the population (which received the highest average returns across its respective $N_{\text{trials}}$ episodes) is selected for evaluation (Ha & Schmidhuber, 2018; Kim et al., 2020). This controller is evaluated for another $N_{\text{pop}} \times N_{\text{trials}}$ episodes in the Dropout's Dream Land environments. The controller's mean across $N_{\text{pop}} \times N_{\text{trials}}$ trials is logged to the Dream Leader Board. After 2000 generations, the controller at the top of the Dream Leader Board is evaluated in the real environment.

### 3.2.4 INTERACTING WITH THE REAL ENVIRONMENT

In Figure 3 we illustrate the controller's interaction with the real environment ($e^*$). The controller only interacts with the target environment during testing. These interactions are never used to modify parameters of the controller. At test time $r$, $d$, and $o$ are generated by the target environment ($e^*$) and $\mathbf{z}$ is the embedding of $o$ from the VAE ($V$). The only use of $M$ when interacting with the target environment is producing $\mathbf{h}$ as a feature for the controller.

Interactions with $e^*$ do not apply dropout to the input/hidden units of $M$. The purpose of applying dropout when training (Sections 3.2.1 and 3.2.2) and selecting the controller (Section 3.2.3) was to

|  | dream | real |
|---|---|---|
| random policy | N/A | $210 \pm 108$ |
| GameGAN | N/A | $765 \pm 482$ |
| Action-LSTM | N/A | $280 \pm 104$ |
| WM | $1465 \pm 633$ | $849 \pm 499$ |
| DDL | $1221 \pm 664$ | $\mathbf{933 \pm 552}$ |

Table 1: Returns from baseline methods and DDL ($p_{\text{train}} = 0.05$ and $p_{\text{infer}} = 0.1$) on the DoomTakeCover-v0 environment.

|  | CarRacingFixedN | | CarRacing-v0 |
|---|---|---|---|
|  | dream | real | |
| random policy | N/A | $-50 \pm 38$ | $-53 \pm 41$ |
| WM | $641 \pm 351$ | $399 \pm 135$ | $388 \pm 157$ |
| DDL | $881 \pm 214$ | $\mathbf{625 \pm 289}$ | $\mathbf{610 \pm 267}$ |

Table 2: Returns from a random policy, World Models and DDL ($p_{\text{train}} = 0.05$ and $p_{\text{infer}} = 0.1$) on the CarRacingFixedN and the original CarRacing-v0 environments.

force the controller to do well on a variety of environments at the cost of dynamics model accuracy (detailed experimental results are in Section 4.2). At the time of interaction with $e^*$ the controller's parameters are frozen and thus there is no benefit to trading dynamics model accuracy.

## 4 EXPERIMENTS

In this section, we perform experiments on the DoomTakeCover-v0 (Paquette, 2017) and CarRacing-v0 (Klimov, 2016) environments from OpenAI Gym (Brockman et al., 2016). These have also been used in related works (Kim et al., 2020; Ha & Schmidhuber, 2018). Architecture details of $V$, $M$, and $C$ are in Appendix A.1.

DoomTakeCover is a control task in which the goal is to dodge fireballs for as long as possible. The controller receives a reward of +1 for every step it is alive. The maximum number of frames is limited to 2100.

CarRacing is a continuous control task to learn from pixels. The race track is split up into "tiles". The goal is to make it all the way around the track (i.e., crossing every tile). We terminate an episode when all tiles are crossed or when the number of steps exceeds 1000. Let $N_{\text{tiles}}$ be the total number of tiles. The simulator (Klimov, 2016) defines the reward $r_t$ at each timestep as $\frac{100}{N_{\text{tiles}}} - 0.1$ if a new tile is crossed, and $-0.1$ otherwise. The number of tiles is not explicitly set by the simulator. We generated 10000 tracks and observed that the number of tiles in the track appears to follow a normal distribution with mean 289. To simplify the reward function, we fix $N_{\text{tiles}}$ to 289 in the randomly generated tracks, and call the modified environment CarRacingFixedN.

For all experiments the controller is trained exclusively in the dream environment (Section 3.2.1) for 2000 generations. The controller only interacts with the target environments for testing (Section 3.2.4). The target environment is never used to update parameters of the controller. Broadly speaking, our experiments are focused on either evaluating the dynamics model ($M$) or the controller ($C$). Accuracy of the dynamics model is evaluated against a training and testing set of trajectories (Appendix A.2). The controller is evaluated by returns in the dream and real environments (Appendix A.2).

### 4.1 COMPARISON WITH BASELINES

We compare the controller trained in Dropout's Dream Land (DDL) with the baseline World Models (WM) controller and a random policy. On the Doom environment we also compare with GameGAN (Kim et al., 2020) and Action-LSTM (Chiappa et al., 2017)[2]. All controllers are trained entirely in dream environments. Results on the dream and real environments are in Tables 1 and 2. The CarRacing-v0 results appear different from those found in World Models (Ha & Schmidhuber, 2018) because we are not performing the same experiment. In this paper we train the controller entirely in the dream environment and only interact with the real environment during testing. In World Models (Ha & Schmidhuber, 2018) the controller was trained directly in the CarRacing-v0 environment.

In Table 1 we observe that DDL offers performance improvements over all the baseline approaches. The DoomTakeCover-v0 returns from DDL are lower than the returns reported by the temperature-regulated variant in Ha & Schmidhuber (2018). Even though the temperature-regulated variant increases uncertainty of the dream environment it is still only capable of creating *one* dream environment. Furthermore, we emphasize that adjusting temperature is only useful for a limited set of dynamics models. For example, it would not be straightforward to apply temperature to any dynamics model which does not produce a probability density function (ex: GameGAN); whereas

---

[2]Results on GameGAN and Action-LSTM returns are from Kim et al. (2020)

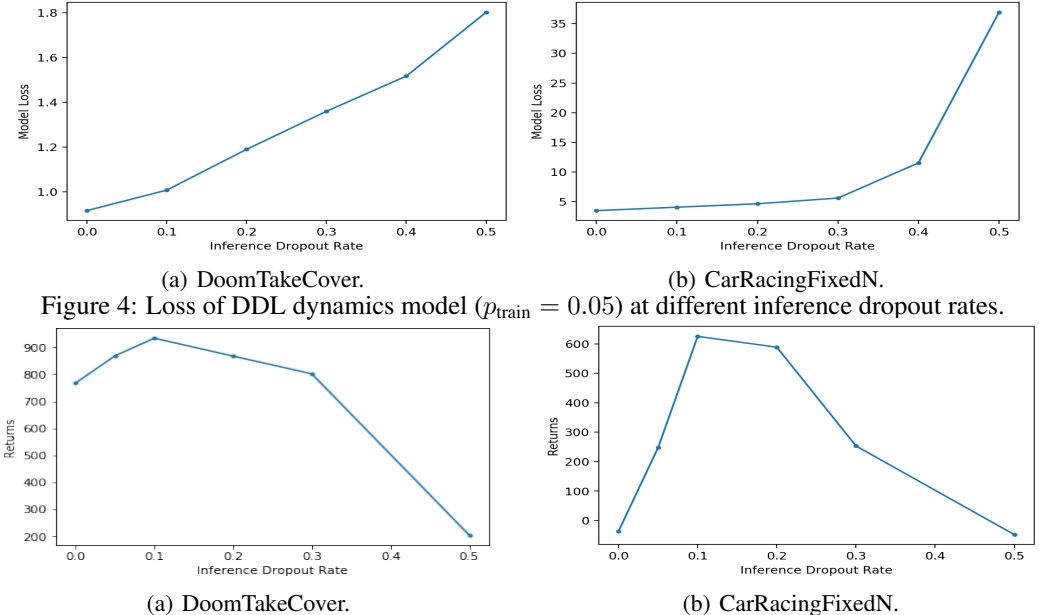

(a) DoomTakeCover.                   (b) CarRacingFixedN.

Figure 4: Loss of DDL dynamics model ($p_{\text{train}} = 0.05$) at different inference dropout rates.

(a) DoomTakeCover.                   (b) CarRacingFixedN.

Figure 5: DDL ($p_{\text{train}} = 0.05$) returns at different inference dropout rates in the real environments.

the DDL approach of generating many *different* dynamics models is useful to any learned neural network dynamics model.

In Table 2 we observe that DDL achieves a significant performance boost over WM in both the real CarRacing-v0 and CarRacingFixedN environment variants. This is because the WM dream environment was easier for the controller to exploit and overfit to errors between the simulator and reality. Forcing the controller to succeed in many different dropout environments makes it difficult to overfit to discrepancies between the dream environment and reality.

## 4.2 DOES DROPOUT AFFECT ACCURACY OF THE RNN?

In this experiment, we compare the losses in (5) achieved by a standard RNN and an RNN with dropout ($p_{\text{train}} = 0.05$). The same training and test sets described in Appendix A.2 are used.

Standard use cases of dropout generally observe a larger training loss but lower test loss relative to the same model trained without dropout (Srivastava et al., 2014; Gal & Ghahramani, 2016a). In Table 3, we do not observe any immediate performance improvements of the RNN trained with dropout. In fact, we observe worse results on the test set. It is possible that the combination of the RNN's small size and dropout are causing the dynamics model to underfit.

The poor performance of both DDL RNNs indicates a clear conclusion about the results from Tables 1 and 2. The improved performance of DDL relative to World Models comes from forcing the controller to operate in many different environments and not from a single more accurate dynamics model $M$.

|  | DoomTakeCover | | CarRacingFixedN | |
| --- | --- | --- | --- | --- |
|  | training loss | test loss | training loss | test loss |
| without dropout | 0.89 | **0.91** | 2.36 | **3.10** |
| with dropout | 0.93 | **0.91** | 3.19 | 3.57 |

Table 3: RNN's loss with and without dropout ($p_{\text{train}} = 0.05$ and $p_{\text{infer}} = 0$) during training.

## 4.3 DOES THE INFERENCE DROPOUT RATE AFFECT DREAM2REAL GENERALIZATION?

It is uncommon to evaluate test loss while still applying dropout. In Dropout's Dream Land this is important because we **still** apply dropout at inference time of the RNN. In this experiment we evaluate the relationship between the inference dropout rate ($p_{\text{infer}}$), model loss, and returns in the target environment. Model loss is measured by the loss in (5) on the test sets for varying levels of inference dropout ($p_{\text{infer}}$). Returns in the target environment are reported based on the best controller (Section 3.2.3) trained with varying levels of inference dropout.

Model loss results are shown in Figure 4. As expected, as the inference dropout rate is increased our model loss increases. However, we also observe in Figure 5 that increasing the inference dropout

rate improves generalization to the target environment. We believe that the increase in returns comes from an increase in capacity to distort the dynamics model. Figures 4 and 5 suggest that we can trade accuracy of the dream environment for better generalization on the target environment. However, this should only be useful up to the point where *the task at hand is fundamentally changed*. Figure 5 suggests this point is somewhere between $0.1$ and $0.2$ for $p_{\text{infer}}$, though we suspect in practice this will be highly dependent on network architecture and the environment.

In Figure 5 we observe relatively weak returns on the real CarRacingFixedN environment when the inference dropout rate is zero. Recall from Table 3 that the dropout variant has a much higher test loss than the non-dropout variant on CarRacingFixedN. This means that when $p_{\text{infer}} = 0$ the *single* environment DDL is able to create is relatively inaccurate. It is easier for the controller to exploit any discrepancies between the dream environment and target environment because only a single dream environment exists. However, as we increase the inference dropout rate it becomes harder for the controller to exploit the dynamics model, suggesting that DDL is especially useful when it is difficult to learn an accurate World Model.

## 4.4 WHEN SHOULD DROPOUT MASKS BE RANDOMIZED DURING CONTROLLER TRAINING?

In this ablation study we evaluate when the dropout mask should be randomized during training of $C$. We consider two possible approaches of when to randomize the masks. The first case only randomizes the mask at the beginning of an episode (*episode randomization*). The second case samples a new dropout mask at every step (*step randomization*). We also consider if it is effective to only apply dropout at inference time but not during RNN training (i.e., $p_{\text{infer}} > 0, p_{\text{train}} = 0$).

Table 4 shows the results. As can be seen, randomizing the mask at each step offers better returns on both real environments. Better returns in the real environment when applying step randomization comes from the fact that the controller is exposed to a much larger number ($> 1000\times$) of dream environments. We also observe that applying step randomization without training the dynamics model with dropout yields a weak policy on the real environment. This is due to the randomization fundamentally changing the task. Training the RNN with dropout ensures that at inference time the masked RNN is meaningful.

| | DoomTakeCover | | CarRacingFixedN | |
|---|---|---|---|---|
| | dream | real | dream | real |
| episode randomization ($p_{\text{train}} = 0.05$, $p_{\text{infer}} = 0.1$) | $505 \pm 460$ | $786 \pm 469$ | $346 \pm 287$ | $601 \pm 197$ |
| step randomization ($p_{\text{train}} = 0.05$, $p_{\text{infer}} = 0.1$) | $1221 \pm 664$ | $\mathbf{933 \pm 552}$ | $881 \pm 214$ | $\mathbf{625 \pm 289}$ |
| step randomization ($p_{\text{train}} = 0$, $p_{\text{infer}} = 0.1$) | $320 \pm 185$ | $339 \pm 90$ | $331 \pm 256$ | $-43 \pm 52$ |

Table 4: Returns of the controller with different frequencies to randomize the dropout mask.

## 4.5 DOES THE NUMBER OF DREAM ENVIRONMENTS AFFECT RETURNS?

Motivated by a similar experiment from Cobbe et al. (2019), we measure the relationship between the number of inference dropout masks available to the dream environments and returns of the controller in the dream and real CarRacingFixedN environments. Limiting the number of masks is the same as controlling for the number of possible environments.

At the beginning of controller training we sample $m$ sets[3] of masks according to $p_{\text{infer}} = 0.1$, yielding $m$ dream environments. While optimizing the controller (Sections 3.2.2 and 3.2.3) we randomize the dream environment at every step by uniformly sampling from the $m$ possi-

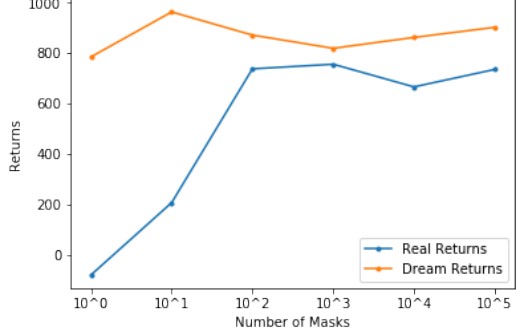

Figure 6: Returns achieved by the controller when varying the number of dropout masks in the real and dream CarRacingFixedN environment.

ble dream environments ($\hat{e}^j$ where $0 \le j < m$). Increasing $m$ forces the controller to be capable of maximizing expected returns across a larger number of dream environments.

As can be seen from Figure 6, interestingly, a relatively small number of masks offers substantial improvements to the controller's ability to generalize to the target environment. We also notice that

---

[3]Each set has $\mathbf{m}_{xi}, \mathbf{m}_{xf}, \mathbf{m}_{xw}, \mathbf{m}_{xo}, \mathbf{m}_{hi}, \mathbf{m}_{hf}, \mathbf{m}_{hw}$, and $\mathbf{m}_{ho}$.

using more masks does not necessarily imply better generalization. This suggests that there exists a smarter way to sample masks, aligned with similar observations in domain randomization (Mehta et al., 2020).

## 4.6 IS DROPOUT'S DREAM LAND MORE THAN A REGULARIZER?

In this experiment we compare Dropout's Dream Land with three regularization methods. First, we consider applying the standard use case of dropout ($0 < p_{\text{train}} < 1$ and $p_{\text{infer}} = 0$). Second, we consider a noisy variant of $M$, where every $z$ adds a small amount of noise ($\epsilon \sim \mathcal{N}(0, 10^{-8})$). Third, we apply L2 regularization to the weights (with L2 penalty $\lambda = 0.001$) during the training process of $M$.

In Table 5, we do not observe any of the standard regularization methods (L2, Dropout, Noise) to improve World Models generalization from dream environments to target environments. L2 Regularized World Models and Dropout World Models can both be viewed as regularizers on $M$. Noisy World Models can be viewed as a regularizer on the controller $C$. The strong returns on the real environment by DDL suggest that it is doing more than standard regularization techniques.

|  | CarRacingFixedN | | CarRacing-v0 |
|---|---|---|---|
|  | dream | real |  |
| World Models | $641 \pm 351$ | $399 \pm 135$ | $388 \pm 157$ |
| L2 Regularized World Models | $760 \pm 419$ | $-76 \pm 8$ | $-90 \pm 1$ |
| Dropout World Models | $1585 \pm 268$ | $-36 \pm 19$ | $-36 \pm 20$ |
| Noisy World Models | $639 \pm 350$ | $277 \pm 135$ | $270 \pm 167$ |
| Dropout's Dream Land | $881 \pm 214$ | $\mathbf{625 \pm 289}$ | $\mathbf{610 \pm 267}$ |

Table 5: Returns from World Models, L2-Regularized World Models ($\lambda = 0.001$), Dropout World Models ($p_{\text{train}} = 0.05$ and $p_{\text{infer}} = 0.0$), Noisy ($\mathcal{N}(0, 10^{-8})$) World Models, and DDL ($p_{\text{train}} = 0.05$ and $p_{\text{infer}} = 0.1$) on the CarRacingFixedN and the original CarRacing-v0 environments.

## 4.7 COMPARISON TO EXPLICIT ENSEMBLE METHODS

In this experiment we compare Dropout's Dream Land with two other approaches for randomizing the dynamics of the dream environment. We consider using an explicit ensemble of a population of dynamics models. Each environment in the population was trained with a different initialization and different mini-batches. With the population of World Models we train a controller with Step Randomization and a controller with Episode Randomization. Note that the training cost of dynamics models and RAM requirements at inference time scale linearly with the population size. Due to the large computational cost we consider a population size of 2.

In Table 6, we observe that neither Population World Models (PWM) Step Randomization or Episode Randomization help close the Dream2Sim gap. Episode Randomization does not help because the controller is forced to understand the hidden state (**h**) representation of every $M$ in the population. Step Randomization performs even worse than Episode Randomization because ontop of the previously state limitations, each of the dynamics models in the population is also forced to be compatible with the hidden state (**h**) representation of all other dynamics models in the population. Even though we use a modest population size, PWM is plagued with issues that will not be fixed by a larger population size. DDL does not suffer from any of the previously stated issues and is also computationally cheaper because only one $M$ must be trained as opposed to an entire population.

|  | CarRacingFixedN | | CarRacing-v0 |
|---|---|---|---|
|  | dream | real |  |
| World Models | $641 \pm 351$ | $399 \pm 135$ | $388 \pm 157$ |
| PWM Episode Randomization | $724 \pm 491$ | $398 \pm 126$ | $402 \pm 142$ |
| PWM Step Randomization | $10 \pm 20$ | $-78 \pm 14$ | $-77 \pm 13$ |
| Dropout's Dream Land | $881 \pm 214$ | $\mathbf{625 \pm 289}$ | $\mathbf{610 \pm 267}$ |

Table 6: Returns from Population World Models (PWM) Episode Randomization, PWM Step Randomization, Noisy World Models, and DDL ($p_{\text{train}} = 0.05$ and $p_{\text{infer}} = 0.1$) on the CarRacingFixedN and the original CarRacing-v0 environments.

## 5 CONCLUSION

In this work we introduce an approach to improve controller generalization from dream environments to reality at little cost. To the best of our knowledge this is the first work to bridge the reality gap between learned simulators and reality. Future direction for this work could be modifying the dynamics model parameters in a targeted manner (Wang et al., 2019; 2020; Such et al., 2019). This simple approach to generating different versions of a model could also be useful in committee-based methods (Settles, 2009; Sekar et al., 2020).

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

## A  Appendix

### A.1  Architecture Details

We follow the same architecture setup as World Models. We adopt the following notation (Kim et al., 2020) to describe the VAE architecture.

**Conv2D(a, b, c)**: 2D-Convolution layer with output channel size **a**, kernel size **b**, and stride **c**. All use valid padding and relu activations.

**T.Conv2D(a, b, c)**: Transposed 2D-Convolution layer with output channel size **a**, kernel size **b**, stride **c**. The final layer uses a sigmoid activation but every other layer uses relu activations.

**LSTM(a)**: LSTM layer with **a** units.

**Dense(a)**: Fully Connected layer with output size **a** followed by a relu activation.

**Linear(a)**: Linear layer with output size **a**.

**Reshape(a)**: Reshape input to output size **a**.

| DoomTakeCover | CarRacing |
|---|---|
| Conv2D(32, 4, 2) | Conv2D(32, 4, 2) |
| Conv2D(64, 4, 2) | Conv2D(64, 4, 2) |
| Conv2D(128, 4, 2) | Conv2D(128, 4, 2) |
| Conv2D(256, 4, 2) | Conv2D(256, 4, 2) |
| Reshape(1024) | Reshape(1024) |
| Linear(32), Linear(32) | Linear(64), Linear(64) |
| Dense(1024) | Dense(1024) |
| Reshape(1, 1, 1024) | Reshape(1, 1, 1024) |
| T.Conv2D(128, 5, 2) | T.Conv2D(128, 5, 2) |
| T.Conv2D(64, 5, 2) | T.Conv2D(64, 5, 2) |
| T.Conv2D(32, 6, 2) | T.Conv2D(32, 6, 2) |
| T.Conv2D(3, 6, 2) | T.Conv2D(3, 6, 2) |

Table 7: Architecture for VAE ($V$) in DoomTakeCover and CarRacing.

| DoomTakeCover | CarRacing |
|---|---|
| LSTM(512) | LSTM(256) |
| Dense(961) | Dense(482) |

Table 8: Architecture for the dynamics model ($M$) in DoomTakeCover and CarRacing.

| DoomTakeCover | CarRacing |
|---|---|
| Linear(1) | Linear(3) |

Table 9: Architecture for the controller ($C$) in DoomTakeCover and CarRacing.

### A.2  Environment Details

Means and standard deviations of returns achieved by the best controller (Section 3.2.3) in the target environment (Section 3.2.4) are reported based on 100 trials for CarRacing and 1000 trials for Doom-TakeCover.[4] Returns in the dream environment are reported based on 1024 trials (Section 3.2.3) for both CarRacing and DoomTakeCover.

**DoomTakeCover Environment** For all tasks on this environment, we collect a training set of 10000 trajectories and a test set of 100 trajectories. A trajectory is a sequence of state ($\mathbf{z}$), action ($\mathbf{a}$), reward ($r$), and termination ($d$) tuples. Both datasets are generated according to a random policy. Following the same convention as World Models (Ha & Schmidhuber, 2018), on the DoomTakeCover environment we concatenate $\mathbf{z}$, $\mathbf{h}$, and $\mathbf{c}$ as input to the controller. In (5), we set $\alpha_d = 1$ and $\alpha_r = 0$ because the Doom reward function is determined entirely based off whether the controller lives or dies.

---

[4]100 trials are used for the baselines GameGAN and Action-LSTM.

**CarRacing Environment** For all tasks on this environment, the training set contains $5000$ trajectories and the test set contains $100$ trajectories. Both datasets are collected by following an expert policy with probability $0.9$, and a random policy with probability $0.1$. The expert policy was trained directly on the CarRacing-v0 environment and received an average return of $885 \pm 63$ across $100$ trials. In comparison, the performance of the random policy is $-53 \pm 41$. This is similar to the setup in GameGAN (Kim et al., 2020) on the Pacman environment which also used an expert policy. For this environment, we set $\alpha_d = \alpha_r = 1$ in (5).

