# OpenReview forum: "Dropout's Dream Land: Generalization from Learned Simulators to Reality"
_ICLR.cc/2021/Conference — Reject_

### Official Review · AnonReviewer3 · 2020-10-19
**Official Blind Review #3**

**Rating:** 6
**Confidence:** 4

**Review:**

##########################################################################

**Summary**:

This paper presents a novel idea of using Dropout in the learned dynamics model for domain randomization. While domain randomization has been extremely popular in the field of sim-to-real transfer, it has been relatively less explored when the dynamics model itself is learned. The paper applies Dropout on the recurrent dynamics model (LSTM) and shows that it helps narrow the reality gap between the imagined (dream) environments and the reality.  Overall, I find the idea to be intuitive and simple. The paper also provides good results in a few environments. But the environments used in the paper are not strong enough to support the claim that using Dropout on the learned dynamics model can bridge the reality gap as all the environments are in simulation.

##########################################################################

**Strengths**:

The idea of applying Dropout on a learned dynamics model is new. The paper is mainly built upon the Recurrent World Model [1]. And it is simple to implement the idea in the Recurrent World Model.

The paper is well motivated, and it is clear that domain randomization requires extra efforts to be applied on a learned dynamics model, and Dropout is one method to incorporate domain randomization into the recurrent dynamics model.


##########################################################################

**Weaknesses**:

The paper claims that adding Dropout to the dynamics model can bridge the reality gap. However, both environments used in the paper are simulated video game environments. The paper does not really show the capability to narrow the sim-to-**real** gap but rather the sim-to-**sim** gap. It remains unclear whether such an approach will improve the performance of transferring a policy trained in simulation to the real world (for example, transfer a navigation skill from a simulated agent to the real robot).


Suggestions on more baselines:
* A baseline where other regularization techniques (such as weight decay, entropy maximization) are used to prevent the dynamics model M from being overfitted. In section 4.2, the paper shows that adding Dropout leads to higher training and testing loss for the dynamics model `M`, and argues that the reason why adding Dropout helps improve the final performance is that it forces the controller to operate in many different environments created by a less accurate `M`. If we use other regularization techniques such as adding weight decay and maximizing the entropy of the distribution to make sure the dynamics model `M` is underfitted or less overfitted, does the final performance also get improved?

* Another way to add randomness to the dynamics model is adding noise to the prediction. Even though the paper uses a Gaussian Mixture, it could be that after supervised training, the variance of each component becomes small without Dropout. So one can add noise to the output to add more randomness to the prediction. For example, suppose the dynamics model makes a prediction `z`, we can either add a gaussian noise `\delta` on top of it to get the output `z_o` (`z_o=z+\delta`) or interpolate between `z` and a uniformly sampled vector `z_u` (`z_o=\lambda * z+ (1-\lambda) * z_u`).



Questions:

In Section 2.2, “This is then repeated for each of the n features”. `n` is the dimension of the latent vector `z`. So do you sample a component from the Gaussian mixture for each dimension separately? Why not sample a component and then sample a vector `z` from the selected multivariate Gaussian distribution directly?

In Figure 2: should there be an arrow that goes from `z` into `M` as the dynamics model `M` takes as input the latent state `z` and action `a`.



##########################################################################

**Minor points**:

Please make Figure 6 more legible.

It would provide more insights if the paper can provide some visualization on diverse simulated trajectories generated from the recurrent model. For example, start from a state `o_0`, apply the same sequence of actions `a_0, a_1, …, a_n`, then use the recurrent model with dropout to generate many sequences of future observations (reconstructed by the VAE decoder). We would expect using the dropout will yield many visually different trajectories with the same initial state and action sequence.

[1] Ha, David, and Jürgen Schmidhuber. "Recurrent world models facilitate policy evolution." Advances in Neural Information Processing Systems. 2018.

---

> ### Author Response · Authors · 2020-11-24
> **Response to AnonReviewer3**
>
> Thank you for your questions and suggestions, we have included our inline response below.
>
> ## "The paper does not really show the capability to narrow the sim-to-real gap but rather the sim-to-sim"
>
> In the camera-ready version we will update the paper to reflect that our experiments only verify that DDL is effective at crossing the dream2sim gap.
>
> ## "Suggestions on more baselines"
>
> As per your suggestions we have introduced three additional baseline comparisons. First, we apply standard dropout as a regularizer on $M$. Second, we introduce a Noisy World Models which adds a small amount of gaussian noise to $\mathbf{z}$ at each step. Third, we apply L2 regularization to the weights during the training process of $M$. We did not find any one of these approaches to benefit as much as DDL. We have attached the table below and included it in the updated submission (Section 4.6).
>
> | | Dream CarRacingFixedN-v0 | Real CarRacingFixedN-v0 | Real CarRacing-v0 |
> |-----|-----|----|-----|
> | World Models | $641 \pm 351$ | $399 \pm 135$ | $388 \pm 157 $ |
> | Dropout World Models | $1585 \pm 268$ | $-36 \pm 19$ | $-36 \pm 20$ |
> | Noisy World Models ($\mathcal{N}(0, 10^{-8})$) | $639 \pm 350$ | $277 \pm 135$ | $270 \pm 167$ |
> | L2 ($\lambda=0.001$) Regularized World Models | $760 \pm 419$ | $-76 \pm 8$ | $-90 \pm 1$ |
> | DDL | $ 881 \pm 214 $ | $\mathbf{625 \pm 289}$ | $\mathbf{610 \pm 267}$ |
>
> ## "In Section 2.2, 'This is then repeated for each of the n features'. n is the dimension of the latent vector z. So do you sample a component from the Gaussian mixture for each dimension separately? Why not sample a component and then sample a vector z from the selected multivariate Gaussian distribution directly?"
>
> Correct, each dimension of the feature has its own corresponding gaussian mixture. We adopted the same approach used by World Models [1].
>
> ## "In Figure 2: should there be an arrow that goes from z into M as the dynamics model M takes as input the latent state z and action a."
>
> We have updated Figures 2 & 3 to include an arrow for $z$ and $h$ going back to $M$ in the camera-ready version.
>
> [1] Ha, David, and Jürgen Schmidhuber. "Recurrent world models facilitate policy evolution." Advances in Neural Information Processing Systems. 2018.

---

> > ### Comment · AnonReviewer3 · 2020-11-24
> > **Why is the variance of noisy world models so small?**
> >
> > Thanks for your response. Can you clarify why do you choose such a small variance ($N(0, 10^{-8})$) for the noisy world models? Isn't that small enough that does not really add any noise?

---

> > > ### Author Response · Authors · 2020-11-25
> > > **Response to AnonReviewer3**
> > >
> > > Indeed it does seem like a small amount of noise. However, even this small amount of noise made the controller significantly worse at generalizing to the real environment. Keep in mind that the $M$ and $V$ parameters of Noisy World Models are identical to the $M$ and $V$ parameters of World Models. If small amounts of noise yield poor results (worse than no noise) we don't expect larger values of noise to perform better. We felt our resources would be better spent on different additional baselines (ex: explicit ensembles, L2 regularization). We are currently running an expirment with larger noise but it will not finish before the rebuttal deadline. In the camera-ready version we will include additional noisy baseline.

---

### Official Review · AnonReviewer1 · 2020-10-29
**Interesting but major concerns with the evaluation**

**Rating:** 4
**Confidence:** 3

**Review:**

The paper introduces the use of dropout for World Models. The argument put forward is that different dropout masks essentially lead to different “dream” environments. The authors compare their approach to the original World Models paper and the recent Game GAN work in the two original World Model environments (Doom and Car Racing). The authors appear to show results that outperform these baselines, and they then perform ablation studies to fully dig into the implications of this application of dropout.

This is a relatively simple approach, but that is not necessarily a bad thing and has the potential to be quite valuable. The evaluation shows positive results (though I have concerns I’ll get to in a moment), and the ablation studies offer valuable insights.

I am concerned with the authors evaluation and the baseline performance. From the original World Models paper Ha and Schmidhuber’s approach achieved an average return of 1092 +/- 556, compared to the value reported in this paper as 849 +/- 499. Similarly for Car Racing the original World Models paper reports 906 +/- 21 whereas this paper reports 388 +/- 157. Both of the scores reported by the World Models paper notably outperform the dropout approach from this paper. It is unclear why this is the case. Potentially the authors of this paper may have used a different optimization method besides CMA-ES. Regardless, it is concerning and casts a shadow on the most important part of this paper.

I am willing to be convinced otherwise but based on the evaluation potentially having major flaws and this being the crux of the paper, I lean towards rejecting it.

Questions:
-There’s a claim of the size of possible different environments in section 3.2.2, but how different are each of these environments from one another? A visualization or some summarizing statistics would be helpful here.
-I don’t understand the leaderboard structure used in the evaluation or why it was necessary, could the authors explain this?
-It may be obvious from up above, but can the authors account for the discrepancy in the performance of the original World Models approach?

Small note: There’s a couple places in the paper with odd language. The final sentence of the abstract is a bit awkward and the beginning of the related works section rapidly changes tenses when describing prior work. Overall though the paper is very well-written.

---

> ### Author Response · Authors · 2020-11-24
> **Response to AnonReviewer1**
>
> Thank you for your review, we have included an inline response below.
>
> ## DoomTakeCover results
>
> The DoomTakeCover-v0 results are generated based off of our implementation of World Models. The results we report are based on when temperature is not regulated ($\tau=1.0$). In the original World Models paper they report similar results when temperature is not adjusted ($868 \pm 511$ [1]). The score of $1092 \pm 556$ they report based on $\tau=1.15$. We do not use the temperature variant for two reasons.  The first, modifying temperature is only applicable when a PDF of the next latent-state is available (ex: not GameGAN [2]). The second, we were unable to reproduce the results from the temperature regulated variant. We could not reproduce the results with the [original implementation](https://github.com/hardmaru/WorldModelsExperiments/tree/master/doomrnn) or ours. Here is a link to an [anonymized repo](https://github.com/lucid-galileo/WorldModels) including a [logfile](https://github.com/lucid-galileo/WorldModels/blob/master/WorldModelsExperiments/doomrnn/tau1.15_eval.txt) from evaluation.
>
> ## CarRacing results
>
> The CarRacing-v0 results appear different from those found in World Models because we are not performing the same experiment. In this paper we train the controller entirely in the dream environment and only interact with the real environment during testing. In the World Models paper the controller is trained directly in the CarRacing-v0 environment.
>
> ## "There’s a claim of the size of possible different environments in section 3.2.2, but how different are each of these environments from one another? A visualization or some summarizing statistics would be helpful here."
>
> In the camera-ready version we will include some qualitative visualizations in the appendix.
>
> ## "I don’t understand the leaderboard structure used in the evaluation or why it was necessary, could the authors explain this?"
>
> The purpose of the leaderboard structure is to prevent the CMA-ES optimization process from regressing. The same approach is used in WM and the GameGAN [2] CMA-ES controller, though neither of the papers gave a detailed explanation. We feel that the leaderboard details are an important foundation to build upon in future research crossing the simulation gap between dream environments and target environments.
>
> [1] Ha, David, and Jürgen Schmidhuber. "Recurrent world models facilitate policy evolution." Advances in Neural Information Processing Systems. 2018.
>
> [2] Kim, Seung Wook, et al. "Learning to Simulate Dynamic Environments with GameGAN." Proceedings of the IEEE/CVF Conference on Computer Vision and Pattern Recognition. 2020.

---

### Official Review · AnonReviewer4 · 2020-10-29
**ICLR 2021**

**Rating:** 6
**Confidence:** 3

**Review:**


# Summary:

The paper builds on the "World Models" (Ha & Shmidhuber 2018) framework with ideas inspired by Domain Randomization (Tobin et. al. 2017 and others).

# Strengths:

 - I find the problem well motivated and clear.
 - The solution is intuitive and simple

# Weaknesses:

 - Overall I find the contribution over the original World Models work relatively minor. This strikes me as little more than World Models + MC Dropout (Gal and Ghahramani 2015) in the VAE. Perhaps I'm missing something about why this is challenging or difficult.

 - Related to the above, there are few statements that I find I bit puzzling. There is a high emphasis put on the fact that we should also apply dropout at test time. It is argued for example that "DDL’s use of dropout is different from traditional applications of dropout. Generally dropout is only applied at training time but not inference time" and similar things are repeated other places. Application of dropout at test time is exactly MC Dropout (Gal and Ghahramani) and is very commonly done. I agree that randomizing over the model changes the underlying MDP that the RL agent using for learning. This same problem is known in the domain randomization literature.

---

> ### Author Response · Authors · 2020-11-24
> **Response to AnonReviewer4**
>
> Thank you for your review, we have included an inline response below.
>
> ## "This strikes me as little more than World Models + MC Dropout (Gal and Ghahramani 2015) in the VAE. Perhaps I'm missing something"
>
> There are a couple misunderstandings here. Dropout is never applied to the VAE (only the RNN). Furthermore, we do not use MC dropout. MC dropout runs a forward pass multiple times on the same input to generate mean and variance estimates. We only perform the forward pass one time for a given input. The purpose of our approach to dropout is to generate many different versions of the dynamics model. More explicitly, the controller is trained to maximize expected returns across many different dynamics models in the ensemble as opposed to the average of the ensemble.
>
> ## "Application of dropout at test time is exactly MC Dropout (Gal and Ghahramani) and is very commonly done"
>
> We have updated the text (Section 3.2.1) to further emphasize the difference between our use of dropout and MC dropout. MC dropout is used to estimate the mean and variance of the network output which can be used for various downstream tasks. However, during MC dropout none of the individual forward passes are used for anything except approximating the mean and variance. DDL does NOT use MC dropout because we do not use uncertainty estimates and we are not interested in the average output of the ensemble. We apply dropout (to the LSTM $M$) at inference time to generate \textit{different} versions of the dynamics model (the LSTM $M$). Each of dynamics models correspond to a unique dream environment which the controller can be trained in.

---

### Official Review · AnonReviewer2 · 2020-10-31
**Official Blind Review #2**

**Rating:** 3
**Confidence:** 4

**Review:**

The paper proposes a method for improving the generalisation of model-based  RL algorithms. The paper proposes that one should learn a distribution of transition models, which they do by training with Dropout to improve generalization in model-based RL. While the paper is addressing an important problem, sadly it lacks in novelty, it is unclear if the results are significant and there is a lack of discussion and comparison to related work in this direction.

Questions and comments:
1. There is a series of work in model-based RL focused on ensemble methods which leverage a population of transition models throughout training to improve generalization. For example:
  * [Kurutach et al., 2017](https://arxiv.org/abs/1802.10592)  which trains a model-free algorithm on purely imaginary data and uses an ensemble of transition models to avoid overfitting to errors made by the individual models.
  * [Buckman et al. 2018](https://arxiv.org/abs/1807.01675) which trains an ensemble of transition models to capture the uncertainty over next step predictions.
2. Work by [Kahn et. al., 2017](https://arxiv.org/abs/1702.01182) seems to be also using dropout as an approximation to an ensemble of transition models to reduce overfitting in real-world navigation tasks. It would be helpful to discuss how your work differs from this work?
3. It is quite unclear why the latest results of [Ha & Schmidhuber, 2018](https://arxiv.org/abs/1809.01999) with adjusted temperature weren’t used as baseline. They are better across the board, and the arguments provided in Section 4.1 do not seem highly convincing. Did you try to combine your method with a temperature-variant of World Model?
4. While the effects of the dropout probability during inference has been studied in the paper, there is no discussion of why the dropout probability during training has been fixed to 0.05 in all the experiments. Was this as a result of a hyperparameter tuning?
5. In Ablation 4.4, it appears that step-based dropout is performing better than fixed episode dropout. Why wasn’t step-based dropout selected for the rest of the results? Also, doesn’t this suggest that the original motivation for using dropout as a way to leverage an ensemble of transition models isn’t strictly true and instead dropout is doing some kind of regularisation here instead?

Overall, the problem being addressed by the paper is interesting, however, the contribution isn’t particularly novel. At the same time, the experiments are slightly lacking and it is hard to assess the significance of the results, for example seed variability seems high when/if present (it is actually unclear what the reported variability corresponds to in Table 1 & 2) and most figures only show a single curve without any error bars.

Hence, I believe this work is not ready for publication at this time. However, I encourage the authors to improve their work by a) discuss the relevant related work e.g. ensemble methods b) discuss differences to [Kahn et. al., 2017](https://arxiv.org/abs/1702.01182),  c) compare results to an explicit model ensemble (e.g. different initialization of the model) d) report results across multiple training runs,  e) explain the choice of hyperparameters and experimental setup & f) improve the clarity of the paper overall.

---

> ### Author Response · Authors · 2020-11-24
> **Response to AnonReviewer2, Part 1/2**
>
> Thank you for your thoughtful questions and suggestions. We have included an inline reply below.
>
> ## "1. series of work in model-based RL focused on ensemble methods which leverage a population of transition models throughout training to improve generalization"
>
> Kurutach et al., 2017 is focused on an iterative version of the problem where the controller being optimized can continually operate in the real environment to collect samples to train $M$.
> Dropout's Dream Land (and the methods we baseline against) are focused on the setting where the controller can only interact with the real environment at test time.  In the updated version, we have referenced Kurutach et al., 2017 as adjacent work in the introduction.
>
> Buckman et al. 2018 is not relevant as they are focused on dynamically adjusting targets based on model uncertainty. While our work is focused on closing the reality gap between the dream environment and the real environment.
>
> ## "2. Work by Kahn et. al., 2017 seems to be also using dropout as an approximation to an ensemble of transition models to reduce overfitting in real-world navigation tasks. It would be helpful to discuss how your work differs from this work?"
>
> Kahn et. al., 2017 uses a combination of MC dropout and bootstrapping to generate uncertainty estimates for their speed-dependent collision cost. DDL does NOT use MC dropout because we do not use uncertainty estimates and we are not interested in the average output of the ensemble. We apply dropout at inference time to generate \textit{different} versions of the dynamics model. Furthermore, we suspect the bagging approach used in Kahn et. al., 2017 would not work as it would require the controller to understand the hidden state ($\mathbf{h}$) of every RNN in the population (see table below).
>
> Perhaps there is a misunderstanding between our novel use case of dropout and MC dropout. We emphasize that DDL does not use MC Dropout. The purpose of DDL's approach to dropout is to maximize the controller's expected returns across many different dynamics models in the ensemble as opposed to maximizing expected returns on the ensemble average.  We have updated Section 3.2.1 to further emphasize that we are not using MC dropout.
>
> ## "3. It is quite unclear why the latest results of Ha & Schmidhuber, 2018 with adjusted temperature weren’t used as baseline."
>
> We did not use the adjusted temperature results from Ha & Schmidhuber, 2018 for two reasons. The first, modifying temperature is only applicable when a PDF of the next latent-state is available (ex: not GameGAN [1]). The second, we were unable to reproduce the results from the temperature regulated variant. We could not reproduce the results with the [original implementation](https://github.com/hardmaru/WorldModelsExperiments/tree/master/doomrnn) or ours. Here is a link to an [anonymized repo](https://github.com/lucid-galileo/WorldModels) including a [logfile](https://github.com/lucid-galileo/WorldModels/blob/master/WorldModelsExperiments/doomrnn/tau1.15_eval.txt) from evaluation.
>
> ## "4. While the effects of the dropout probability during inference has been studied in the paper, there is no discussion of why the dropout probability during training has been fixed to 0.05 in all the experiments. Was this as a result of a hyperparameter tuning?
>
> We did not perform a hyperparameter sweep over $p_\text{train}$ as this would have been very expensive. A hyperparameter sweep over $p_\text{train}$ would also need to consider the corresponding $p_\text{infer}$. For these reasons we focused our resources on experiments that help understand the algorithm.
>
> ## "5. In Ablation 4.4, it appears that step-based dropout is performing better than fixed episode dropout. Why wasn’t step-based dropout selected for the rest of the results?"
>
> Indeed Step Randomization outperforms Episode Randomization. Step Randomization is used in all DDL experiments. Kurutach et al., 2017 also found step randomization to work well.

---

> > ### Author Response · Authors · 2020-11-24
> > **Response to AnonReviewer2, Part 2/2**
> >
> > ## "doesn’t this suggest that the original motivation for using dropout as a way to leverage an ensemble of transition models isn’t strictly true and instead dropout is doing some kind of regularisation here instead?"
> >
> > We have included an additional baseline which suggests DDL is more than a standard regularization technique.  The additional baseline compares DDL with three regularization methods. First, we apply standard dropout as a regularizer on $M$. Second, we introduce a Noisy World Models which adds a small amount of gaussian noise to $\mathbf{z}$ at each step. Third, we apply L2 regularization to the weights during the training process of $M$. We did not find any one of these approaches to benefit as much as DDL. We have attached the table below and included it in the updated submission (Section 4.6).
> >
> > | | Dream CarRacingFixedN-v0 | Real CarRacingFixedN-v0 | Real CarRacing-v0 |
> > |-----|-----|----|-----|
> > | World Models | $641 \pm 351$ | $399 \pm 135$ | $388 \pm 157 $ |
> > | Dropout World Models ($p_\text{train}=0.05$) | $1585 \pm 268$ | $-36 \pm 19$ | $-36 \pm 20$ |
> > | Noisy World Models ($\mathcal{N}(0, 10^{-8})$) | $639 \pm 350$ | $277 \pm 135$ | $270 \pm 167$ |
> > | L2 ($\lambda=0.001$) Regularized World Models | $760 \pm 419$ | $-76 \pm 8$ | $-90 \pm 1$ |
> > | DDL | $ 881 \pm 214 $ | $\mathbf{625 \pm 289}$ | $\mathbf{610 \pm 267}$ |
> >
> > ## "d) report results across multiple training runs"
> >
> > We follow the same (we use more seeds on DoomTakeCover-v0) approach to reporting results as Ha & Schmidhuber, 2018. As mentioned in Appendix A.2, the mean and standard deviation are reported over 100 or 1000 (depending on environment) trials in the real environment.
> >
> > ## "c) compare results to an explicit model ensemble (e.g. different initialization of the model)"
> >
> > We have included an additional experiment (Section 4.7) comparing with two approaches for randomizing with an explicit ensemble (see Table below).  Population World Model (PWM) trains a population (size 2) of models with different initializations and mini-batches (same full dataset of trajectories).  The training cost of dynamics models scales linearly with the population size, and so we are limited to using a small population size during the rebuttal period.  Even though the population size is small, PWM is plagued with issues that will not be fixed by a larger population size. In short, we don't find it useful as it requires the controller to be compatible with the hidden state ($\mathbf{h}$) representation of every World Model in the ensemble.
> >
> > | | Dream CarRacingFixedN-v0 | Real CarRacingFixedN-v0 | Real CarRacing-v0
> > |-----|-----|----|-----|
> > | World Models | $641 \pm 351$ | $399 \pm 135$ | $388 \pm 157 $ |
> > | PWM Episode Randomization | $724 \pm 491$ | $398 \pm 126$ | $402 \pm 142$ |
> > | PWM Step Randomization | $10 \pm 20$ | $-78 \pm 14$ | $-77 \pm 13$ |
> > | Dropout's Dream Land | $ 881 \pm 214 $ | $\mathbf{625 \pm 289}$ | $\mathbf{610 \pm 267}$ |
> >
> > [1] Kim, Seung Wook, et al. "Learning to Simulate Dynamic Environments with GameGAN." Proceedings of the IEEE/CVF Conference on Computer Vision and Pattern Recognition. 2020.

---

### Author Response · Authors · 2020-11-24
**Summary of Revisions**

The helpful comments from reviewers have inspired us to make the following modifications. We updated Section 3.2.1 to emphasize that DDL does not use Monte Carlo dropout. We added a regularization experiment demonstrating that DDL is more than a traditional regularization technique. We added an explicit ensemble experiment which shows DDL does not suffer from any of the issues (run-time and inconsistent hidden state representations) that plague explicit ensemble approaches. We added arrows showing $\mathbf{z}$, $\mathbf{h}$ are passed to $M$ in Figures 2 and 3. We fixed an error where we inadvertently rendered Figure 5b as Figure 5a. Lastly, we added the relevant (Kurutach et al., 2017, Kahn et. al., 2017) requested references.

---

### Decision · Program_Chairs · 2021-01-07
**Final Decision**

**Decision:**

Reject

**Comment:**

Dropout's Dream Land: Generalization from Learned Simulators to Reality

This work explores the use of dropout inside a learned dynamics model, so that when an agent is trained inside an environment generated by this model (rather than the actual environment), the policy learned would do better in the actual environment. In a sense, this is a form of domain randomization in a learned simulator. They show that their approach has better transfer capabilities compared to the baseline World Models method, where an entropy injection via temperature adjustment is used to make transfer more effective, and this is particularly evident in the CarRacing "learn in latent simulation" experiment.

While this work is interesting to me, and I believe it has something to offer to the ICLR community, after reading the reviews and also the detailed author / reviewer discussion (and after understanding and clarifying all of the nuanced points in the experiments), the reviewers and myself believe that this paper needs more work before meeting the bar of ICLR conference acceptance. I believe the author clarified many issues and misunderstandings with the reviewers, and we have made sure the reviewers took that into consideration. Based on the reviews, I have summarized recommendations below to help the authors improve this work. There are 2 dimensions of the work that can be improved:

1) Novelty and Connections with prior works that used dropout in RL

While this work is not using MC dropout (it applies dropout inside the LSTM M), there has been sufficient work (as listed by R2) using MC dropout with RL, and the reviewers' impression that while they may not be exactly the same, it does bring to question of the novelty in this work. It is recommended to discuss not only that the approach is not MC dropout, but also connections with previous work that used MC dropout (or even other forms of dropout) dynamics models to prevent overfitting, and also discuss why this particular approach is needed (or is better, via experiments), over MC dropout, if that can also be used to generate novel dream environments. As R3 mentioned, the idea of applying dropout (of whichever form) on a learned dynamics model *is* new, and this point should be emphasized very clearly to the reader, so I recommend the authors improve the writing to incorporate discussions and relations to previous work (R2 provides a good list), and emphasize what is considered novel in this particular work.

2) Experimental design

The authors show that the proposed method offers a clear advantage over the baseline WM approach for CarRacing, where they show that DDL can do the "train in dream / deploy in real env" transfer much better than the original baseline can. For the Doom experiment, I'm less concerned about the exact score compared to the baseline (as noted by R1), given the high variance of these results, and also the high randomness in the process used to collect data via a random policy, and see that their result is within the margin of statistical error. Just noting the replication effort that went in as a footnote and citing existing results, noting the high variance, should suffice.

What would really improve the work, IMO, is to compare to GameGAN on PacMan environment. Would DDL offer improvements vs GameGAN on PacMan? This will be a strong data point for the proposed method's effectiveness, compared to more trivial tasks such as DoomTakeCover. R3 also brings out a good point that the paper offers better sim-to-sim transfer, rather than sim-to-real transfer. That is another avenue to explore, if this work is to be improved.

The authors have cited PlaNet / Dreamer / SimPle papers, but mentioned that these works don't deal with the issue of reality gap, but I would argue that the iterative learning (data gathering / retraining) aspect of these algorithms is actually one method to address the gap. The tasks studied in these more recent papers, such as DM Control from Pixels, or Atari, have more datapoints, or "leaderboard" participants, using this paper's terminology, so they can also be considered if the authors wish to try DDL to see if this method can help improve the performance of these newer approaches which are based on world models with iterative training and data collection. One can explore whether this approach can lead to better sample efficiency gains, in addition to absolute performance after training, when combined with iterative training in PlaNet / Dreamer type approaches.

Overall, the work in its current form would make a good workshop paper, but I look forward to seeing more work done in the experiments to see better convincing results, in addition to clarifying the writing on related approaches and making contributions / novelty more clear, which I believe will really improve the work for a future submission.